# Toward a Personalized Psychological Counseling Service in Assisted Reproductive Technology Centers: A Qualitative Analysis of Couples’ Needs

**DOI:** 10.3390/jpm13010073

**Published:** 2022-12-29

**Authors:** Giulia Scaravelli, Fabiola Fedele, Roberta Spoletini, Silvia Monaco, Alessia Renzi, Michela Di Trani

**Affiliations:** 1ART Italian National Register, National Centre for Diseases Prevention and Health Promotion, Italian National Health Institute, Viale Regina Elena 299, 00161 Rome, Italy; 2Department of Dynamic and Clinical Psychology and Health Studies, “Sapienza” University of Rome, Via degli Apuli 1, 00185 Rome, Italy

**Keywords:** infertility, ART, emotional needs, psychological support

## Abstract

Infertility may have a very strong emotional impact on individuals, requiring adequate support, but few studies on patients' demands toward psychological support have been conducted. This study aims to explore the emotions related to the infertility and to the Assisted Reproductive Technology (ART) procedure for which patients consider useful a psychological support. A total of 324 women completed a sociodemographic and clinical questionnaire and an open-ended questionnaire on emotional needs for psychological support. The written texts were explored by the Linguistic Inquiry and Word Count (LIWC) programme and linguistic characteristics were related to sociodemographic and anamnestic variables. Specific linguistic features were connected to several individual characteristics. More specifically, differences in linguistic processes emerged comparing women with an age over or under 40 years, women undergoing their first attempts versus more attempts, women undergoing ART with or without gamete donation, and women undergoing ART for male or unknown causes, as well as those undergoing ART for female or both partners’ problems. These differences seem to confirm that older age, more attempts, gamete donation, and ART for unknown or male causes are risk factors that may worsen women's psychological well-being. This study contributes to increase the knowledge about the emotional needs of patients undergoing an ART procedure to develop specific psychological intervention programs.

## 1. Introduction

Although subject to transformations and change in meaning over time, parenting continues to be defined as a key event in an individual's life cycle, an essential milestone for the couple, and an expected and desired event for the extended family at the tri-generational level. Any impediment to the transition to parenthood, first and foremost, a diagnosis of infertility, could result in a developmental shutdown that is not always easy to process and whose consequences can be declined at the individual, couple, and family levels. The World Health Organization [1] estimates that between 48 million couples and 186 million people worldwide are affected by infertility. In industrialized countries, this condition affects 15–20% of couples. This percentage is unfortunately expected to increase for various reasons, but mainly due to increasing maternal age, environmental pollution, food adulteration, and lifestyle [1]. Moreover, male infertility is increasing, and it is responsible for 38% couple’s infertility [2,3].

International literature highlighted that infertility represents a risk factor for worsening mental health, general well-being, and the couple’s satisfaction [4,5]. In particular, the emotional, relational, social, and economic costs of infertility can profoundly affect self-esteem, sense of self-efficacy, perception of social inclusion, and the adjustment of the couple [6,7], thereby increasing feelings of guilt, shame, and anger [8,9].

Moreover, undergoing an Assisted Reproductive Technology (ART) treatment for fertility issues can represent a further element of fatigue and stress for these couples, since medical procedures could be long, demanding, economically burdensome, and invasive [10,11,12], and patients put their lives on hold, no longer feeling in control of their bodies or their life plans [13]. Moreover, often these couples need an in vitro fertilization (IVF) resolutive treatment after having tried all the techniques known in the literature, such as an injection of embryo culture supernatant to the endometrial cavity [14]. ART treatments are also needed in patients who have to preserve their fertility in case of cancer diagnosis or in patients with genetic disease responsible for different systemic conditions, including infertility, needing specific psychological support [15,16,17].

Data also indicate that only about 20% of couples who begin ART treatment achieve pregnancy and only about 14% obtain the long-awaited baby. The advanced age of the couple’s members and the resulting pathologies of the reproductive system often make it necessary to repeat ART attempts, and therefore deal with repeated failures, before the possible positive outcome [2]. The impact of diagnosis and medical procedure, both on psychological, physical and economic dimensions, seems to contribute to the high rates of treatment dropout, with an estimated 30% of couples discontinuing treatments [18,19,20,21,22,23,24]. Additionally, when the treatment is successful, some studies argue that it is important to follow the Neonatal Outcomes and Long-Term Follow-Up of Children Born from ART [25,26,27].

The diagnosis of infertility and ART treatment can have a very strong emotional impact on the person and the couple, which needs adequate support for emotional processing. In this regard, some of the literature has focused on investigating the needs of couples undergoing ART treatment with the aim of giving rise to increasingly patient-centered services (“Patient-Centered Care”) and personalized medicine [24,28,29,30,31,32,33]. In patient-centered and personalized care approach, patients’ specific health needs and desired health outcomes drive all health care professional decisions, with the aim of individualizing design of care and better aligning it to individual patients' situations [34].

In this direction, some studies have shed light on the information regarding the needs of couples facing ART treatment [24,28]. In fact, most infertility patients do not seem to have a deep understanding of the therapeutic treatments and their success rates, which often leads them to develop unrealistic expectations with respect to success and a strong sense of dissatisfaction after treatment failure [29,30,35]. Communication with fertility clinic staff is also very important for patients, who require listening, understanding, empathy, and a clear, nonjudgmental, honest interaction [24,30,31,32,33,36]. Finally, on the organizational level, good accessibility and sufficient comfortable spaces are required [24,30].

Despite the aforementioned research regarding the patients' needs of assistance, studies on patients' demands toward psychological support to help them cope with infertility and its treatment are still very limited. Previous studies showed that only a small proportion of patients undergoing ART treatment seek psychological counseling, and that there is a particularly low rate of request among men [37,38,39]. Nevertheless, it is known that receiving psychological support can promote patient well-being, treatment continuation, and its outcomes, also considering the different male and female attitudes to infertility and the treatments [40,41,42,43,44]. In addition to patient benefits, psychological care in medical centers has also been associated with a reduced burden on the medical team due to perceived greater support, reduced patient-related stress, and increased confidence in their ability to manage their needs, with improved delivery of patient-centered care and patient satisfaction [45,46]. In Italy, a national survey [47] aiming to explore the diffusion and the characteristics of psychological care offered to infertile couples in ART centers showed that only a few couples resort to psychological counseling (between 10% and 20% of couples) and that psychological care is not a fully operational service in Italian ART centers, nor is it incorporated into their routine practice even though the treatments they provide are known to have a strong emotional impact. All of this highlights the need to explore patients’ specific emotional needs and expectations toward psychological care services at ART centers, with the aim of promoting the diffusion of a patient-centered psychological service.

This study aims to explore the linguistic characteristics of texts written by infertile women undergoing ART treatments for fertility problems regarding their emotions related to the infertility and to the ART procedure for which patients consider useful a psychological support. Moreover, we aimed to explore the associations between the linguistic characteristics and several socio-anamnestic dimensions that, according to the international literature, may represent key elements in the emotional organization of this experience. All of this further aligns to the broader aim to increase the knowledge about the emotional needs of patients undergoing the ART procedure and to develop specific and individualized psychological intervention programs.

## 2. Materials and Methods

### 2.1. Procedure and Participants

The work was carried out in accordance with the Code of Ethics of the World Medical Association (Declaration of Helsinki) for experiments involving humans. Ethical approval was granted by the Ethics Committee of the Department of Dynamic and Clinical Psychology and Health Studies, Sapienza University of Rome. Participation was voluntary and all the participants gave their informed consent before completing the research protocol. The research protocol was co-developed by the ART Italian National Register−Italian National Institute of Health and Department of Dynamic and Clinical Psychology and Health Studies−Sapienza University of Rome. Research protocol included the collection of participants’ sociodemographic and anamnestic information, and it also comprised an open-ended questionnaire aimed to explore women’s emotional and psychological needs associated with the experience of infertility and related treatments.

From March 2021 to October 2021, an online questionnaire was sent to 195 active ART centers (both public and private) in Italy according to the data from the Italian National Assisted Reproductive Technology Register and to eight Associations of ART patients asking for its diffusion. A total of 324 patients undergoing ART treatment participated in the study (293 females, 31 males) and completed the online questionnaire. The male participants were excluded from the data analysis since the reduced male participation did not allow a homogenous gender distribution and the generalizability of the results to the male population. In all, 293 women were included in the study and they had a mean age of 38.02 years (SD = 4.71) and a mean of 43.67 months since the beginning of pregnancy attempts (SD = 29.87). Moreover, 93.5% of the women participating in the study (*n* = 274) underwent in vitro fertilization/intracytoplasmic sperm injection (IVF/ICSI, both second level technique based on in vitro fertilization with embryos transfer) whereas 6.5% underwent intrauterine insemination (IUI, a first-level technique). Table 1 shows all the socio-anamnestic characteristics of the sample considered in the present study (see Table 1).

### 2.2. Measures

Sociodemographic questionnaire. A questionnaire aimed to explore sociodemographic and anamnestic variables, such as age, residence, educational level, employment status, months since the beginning of pregnancy attempts, number of previous ART cycles, and infertility cause.

Open-ended questionnaire. This questionnaire aimed to explore women’s emotional and psychological needs associated with the experience of infertility and related treatments that could benefit from psychological support. This questionnaire consisted of the following prompt: “Considering the ART treatment you are undergoing, could you tell us about the emotions related to your infertility experience and its treatment that you feel could benefit from psychological support?” [“Considerando il percorso di PMA che sta effettuando, quali sono le sue emozioni connesse all’esperienza di infertilità e al relativo trattamento che sente potrebbero beneficiare di un supporto psicologico?”]

Linguistic Inquiry and Word Count (LIWC). The LIWC [48] is the most used and best validated approach in studying word use in social psychology and it can allow the detection of individual psychological differences. Many studies show the power of this measure to detect several individual differences: personality patterns [49]; depression [50] and suicide [51]; lying behavior [52]; and the degree of resolution for the undisclosed event [53]. This program can use more than 80 different dictionaries, which are a collection of words that define a particular category. It calculates how many times words from different dictionaries appeared in a text file. In the end, the program reports a set of dictionaries scores for each file. In this study, we used 37 dictionaries in the Italian version (see Table 2).

### 2.3. Statistical Analysis

A statistical analysis was conducted using the Statistical Package for Social Science (SPSS) Version 25 for Windows (IBM, Armonk, NY, United States). Data were reported as frequencies and percentages for discrete variables and as Ms ± SDs for continuous variables. The women who participated in this study were divided in different subgroups according to the specific socio-anamnestic characteristics considered (age under/over 40 years, the first ART attempt versus the second or another attempt, ART with/without gamete donation, and specific cause of infertility). Since most of the linguistic measures did not follow normal distribution, differences between subgroups of women were evaluated with the use of the Mann–Whitney U and Kruskal–Wallis non-parametric tests for independent samples, to compare two subgroups or more than two subgroups, respectively. Furthermore, when the Kruskal–Wallis tests were used the Mann–Whitney tests were performed as post hoc tests to determine the specific differences between the subgroups, and the Bonferroni correction for multiple comparisons was applied. A *p* < 0.05 was considered significant.

## 3. Results

Table 2 shows the M ± SD values for the linguistic characteristics considered in the present investigation.

Table 3 and Table 4 shows the results of the one-way ANOVA run. In regard to the variable age, the group of women who were 40 years or younger showed a greater use of words per sentence, as well as words related to communication process, future actions, exclusive terminology (such as: but, except, without), and achievement, compared to women who were 40 years or older.

Regarding the variable of ART attempts, the group of women undergoing their first attempt showed a greater use of words expressing anxiety and fearful feelings, words related to social and communication processes, leisure activity, domestic context, third-person pronouns, and the verb “to have”, compared to women undergoing their second or further attempt (see Table 3).

Regarding the variable of ART treatment with/without gamete donation, the women undergoing the medical procedure with gamete donation showed a lower word count and words per sentence, as well as a greater use of words related to the negation processes, family (and family members), and death and dying compared to women using their own gamete.

Regarding the specific cause of infertility, several significances emerged for the use of words related to feelings of anger, inhibition, as well as for the use of first-person singular pronouns (see Table 4). Multiple comparison post hoc analysis with the Bonferroni correction showed that the group of women who underwent the ART procedure without a clear cause of infertility reported a greater use of words reflecting feelings of anger, compared to the women who underwent ART for female (U = 23.595; *p* = 0.004), male (U = 21.098; *p* = 0.049) or both partners’ cause (U = 23.330; *p* = 0.044). In regard to the differences in the use of words related to inhibition and block processes, post hoc comparisons did not show significances. However, the group of women who underwent ART for male cause showed a tendency to the significance for a greater use of words related to inhibition and block processes than the groups of women reporting an unexplained or female infertility cause (both a *p* = 0.06). The women who underwent ART for female cause show a greater use of first-person singular pronouns than women with a male (U = 33.320; *p* = 0.035).

## 4. Discussion

Reflecting on the focused psychological support programs, understanding the emotional needs of women undergoing ART treatments is a useful tool to utilize in patient-centered services. In Italy, even though the presence of a psychological service in ART centers is defined by law, only few couples resort to psychological counseling (between 10% and 20% of couples) [47] and no specific guidelines are settled. With the aim of providing useful information to understand women’s psychological requests, we analyzed the linguistic features of their narratives concerning the emotional needs relating to infertility and ART that they felt could benefit from psychological support. We used the LIWC program, which was developed to provide an efficient method for studying the various emotional, cognitive, structural, and process components present in individuals’ written speech samples.

Specific linguistic patterns emerged, linked to sociodemographic and medical characteristics. First, women aged 40 and over displayed additional elements that could determine a risk and stress factor for ART, also considering the low pregnancy rates in this age group: they used fewer references to communication with others, to social intercourses and to leisure activities, fewer verbs in the future tense, and fewer exclusive words (e.g., but, except, without), when compared to women under 40 years of age. Moreover, women over 40 years expressed less negative emotion than women under 40. In general, these features indicated that women who have an age near to the physiological age-related decline in fertility seemed to show a decreased interest in social relationships and free time, and lower ability to imagine their future. Moreover, their thinking process regarding the emotional needs associated with the infertility condition appeared less complex from a cognitive point of view, since the use of exclusive words that are helpful in making distinctions was less represented [48]. Despite these elements, women seem to rarely express their negative emotions. It is possible to hypothesize that their language reflected a defensive process of disinvestment not only from therapies, but also from the idea and desire to have a child, accompanied with a depressive withdrawal from social relationships [51]. This seems to confirm that for women an age above 40 can be considered a risk factor of both medical and psychological negative outcomes. This issue therefore requires the attention of psychologists due to women’s lower emotional and social resources to cope with treatment difficulties. In fact, this finding appears to be in line with the broader literature showing a positive association in infertile couples between psychological disorders and greater age [54,55].

Another feature that appears relevant, in terms of differences in linguistic patterns, is that of a first ART attempt. More specifically, women who start their first ART attempt used more words related to leisure activities and fewer words related to negations and optimism/energy than women who had attempted ART several times before. This could reflect the positive and negative consequences of undergoing several ART treatments. Women with several ART attempts may have elaborated different strategies to face to this long path, as defensive process against painful emotions, using both negation and optimism for the future. In this vision, it is crucial a psychological support for the women in their first attempt and for the long-term, in order to develop positive strategies instead of negative responses.

The analysis revealed also that women who underwent ART procedures without a clear cause of infertility reported more feelings of anger compared to the women who underwent ART for female or both partners’ problems. We can hypothesize that anger needs to be expressed and elaborated, in order not to affect the couple or involve feelings of guilt. The absence of identification of a clear problem may obstacle this process. Moreover, the group of women who underwent ART for male cause showed a greater use of words related to inhibition and block processes, and a greater use of words related to the body and bodily symptoms, than the groups of women reporting an unexplained, female, or both partners’ infertility cause. In this case, a conscious feeling did not emerge, but the thought appeared inhibited, blocked, and the woman made more references to somatization. The attribution of the cause of infertility to the man seems to represent a more difficult condition than not knowing a cause, because anger cannot be expressed (against the partner), perhaps not even though; the emotional process is inhibited. Women who underwent ART for female cause showed a greater use of the first-person singular pronouns. According to the literature [56,57], the recurrent use of the first-person singular can be associated with depressive feelings. This could reflect the feeling of blame and shame often experienced by woman suffering from fertility problems, which could direct aggressive feelings to herself, creating a depressive condition. Regardless of the cause that determines the infertility condition, the woman has to undergo medical treatment to become pregnant and to realize the couple’s desire for parenthood, and it is known that the medical procedure is complex and contains a number of stressful aspects such as daily injections, blood samples, ultrasound, and laparoscopic surgery with also the possibility of the various phases failing. All of this can negatively affect women’s psychological well-being [55].

Furthermore, women who underwent ART with gamete donation used less words, used the second-person plural pronouns fewer times and fewer words related to the body and sex, as well as a greater use of words referring to “death” than women undergoing ART without gamete donation. The association between the words referring to death appears peculiar and worthy of reflection. Gamete donation represents a condition that inserts a new foreign psychic element, which symbolically refers to the death of the couple’s biological parenthood possibility. The feeling of loss of a homologous parenthood must find space in a psychological setting capable of processing the complex emotions and anxieties of death evoked by the donation of the gametes. This could be reflected also in the reduced use of the pronoun “us” that characterized the way couples refer to themselves, and in the reduced presence of sex and bodily words. Physicians and psychologists should be aware that the ART treatment with gamete donation may activate intense and conflicting emotional experiences. This result is in line with the high request for psychological counseling showed by the couples who are undergoing this specific kind of treatment [47].

Several limitations could be identified in this study. First, the number of participants in the present study is one limitation that can be expanded upon in future studies. Considering the ART centers and associations involved, a greater sample size was expected. This may represent a bias for generalizing the results to this specific clinical population. Second, the absence of male participants in this study should be explored further in future studies, since males also have emotional needs related to infertility and ART. Moreover, the rate of women undergoing ART first-level techniques was limited (6.5%) and in future studies this element should be considered. A further limitation is the absence of psychological tests useful for evaluating psychological dimensions that may have specific associations with the linguistic characteristics of women’s narrative (such as alexithymia, defense mechanisms, and the impact of infertility and ART treatments on the quality of life).

## 5. Conclusions

In conclusion, the analysis of the language used in women's narratives provides useful elements for the design of psychological interventions focused on women’s emotional needs, which are more or less conscious. In addition, the characteristics of language, used to determine the contents of the narratives, allow us to understand the psychological processes that characterize the emotional experiences of women. Data showed that sociodemographic and medical variables could characterize the emotional experiences of women and their way of expressing their needs. Considering these differences could permit us to realize personalized psychological interventions focused on specific patients’ cognitive and emotional needs.

## Figures and Tables

**Table 1 jpm-13-00073-t001:** Sociodemographic and Anamnestic Characteristics of the Sample.

Sociodemographic Variables	M	SD
Age	38.02	4.71
Months since the beginning of pregnancy attempts	43.67	29.87
	%	N
Residence		
Northern Italy	53.1	156
Central Italy	24.8	72
Southern Italy	22.1	65
ART attempts		
1	17.7	52
2	21.8	64
3	21.2	62
4	14.3	42
5	7.5	22
6	7.3	21
≥7	10.2	30
Employment status		
Workman/employee	61.8	181
Freelance	27	79
Unemployed	11.2	33
Educational level		
8 years	4.1	12
13 years	35.8	105
18 years	41.3	121
≥18 years	18.8	55
Infertility cause		
Unknown	21.5	63
Female	42	123
Male	21.2	62
Both partners	15.4	45

**Table 2 jpm-13-00073-t002:** Sample mean values of linguistic measures and dictionary examples.

	Mean	SD	Examples
Words Count	44.35	50.31	
Words per sentence	13.65	9.72	
Total first-person pronouns	1.02	2.15	I, we, me
First-person singular	3.55	8.22	I, my, me
First-person plural	0.57	1.98	we, our, us
Second-person pronouns	0.00	0.00	you, you’ll
Third-person pronouns	0.05	0.33	she, their,
Negations	5.13	11.95	no, never, not
Assents	0.25	1.6	yes, OK
Affective-emotional processes	6.40	6.33	ugly, bitter
Positive emotions	0.41	1.38	happy, pretty, good
Optimism and energy	0.65	2.15	pride, win
Negative emotions	2.62	3.95	hate, worthless
Anxiety or fear	0.98	2.75	nervous, afraid
Anger	0.24	1.07	hate, kill
Sadness or depression	0.73	2.15	cry, sad
Cognitive processes	7.54	8.59	know, ought
Causation	0.93	1.78	because, effect
Insight	3.86	8.27	know, consider
Discrepancy	1.34	2.25	should, would
Inhibition	0.42	1.85	block, constrain
Tentative	1.68	2.50	perhaps, guess
Certainty	1.24	3.58	always, never
Social processes	4.63	8.42	talk, us,
Communication	2.21	7.57	talk, share
Friends	0.15	0.67	buddy, coworker
Family	0.37	1.11	mom, brother
Past tense verb	1.51	2.74	were, had
Present tense verb	5.82	7.53	talk, is,
Future tense verb	0.07	0.39	will, might
Inclusive	0.98	1.79	with, and
Exclusive	2.57	3.49	but, except
Achievement	0.59	1.52	goal, win
Leisure activity	0.92	6.44	house, TV
Home	0.20	0.79	kitchen, lawn
Death and dying	0.02	0.39	dead, burial
Sex and sensuality	0.03	1.25	lust, penis
Body states and symptoms	0.39	1.18	ache, heart
To be verb	0.70	2.35	be, is
To have verb	0.87	1.88	have, had

**Table 3 jpm-13-00073-t003:** Significant differences in linguistic dimensions between groups of women with different socio-anamnestic characteristics.

	Women Undergoing the First ART Attempt *n* = 52			Women Undergoing the Second or More ART Attempts *n* = 241			Statistics Mann–Whitney U Test	*p*
	M	SD	M-Rank	M	SD	M-Rank		
Negations	3.93	10.23	122.12	5.39	12.29	152.37	7559.5	0.015
Optimism/energy	0.21	0.69	130.73	0.75	2.33	150.51	7112.0	0.031
Leisure activity	3.56	14.81	161.72	0.34	1.41	143.82	5500.5	0.021
	Women with age ≤ 39 years *n* = 181			Women with age ≥ 40 *n* = 112				
Negative emotions	2.95	4.16	154.22	2.08	3.54	135.33	8829.0	0.047
Social	5.23	9.65	154.49	3.65	5.81	134.89	8779.5	0.049
Communication	2.94	9.40	157.47	0.99	1.21	130.08	8240.5	0.002
Future tense verb	0.12	0.48	150.08	0.01	0.08	142.03	9579.0	0.039
Exclusive	2.93	3.87	154.35	1.98	2.67	135.12	8805.5	0.045
Leisure activities	1.27	8.06	152.02	0.34	1.75	138.89	9227.5	0.031
	ART without gamete donation *n* = 225			ART with gamete donation *n* = 68				
Words count	48.34	53.33	154.42	31.18	35.96	122.43	5979.5	0.006
Words per sentence	14.48	10.27	153.59	10.92	6.99	125.19	6167.0	0.015
Sex and sensuality	0.35	1.42	150.44	0.03	0.21	135.61	6875.5	0.018
First-person plural	0.62	2.04	150.63	0.37	2.81	134.98	6832.5	0.035
Body states and symptoms	0.46	1.28	151.01	0.16	0.71	133.72	6747.0	0.022
Death and dying	0.000	0.000	146.0	0.10	0.80	150.31	6875.5	0.010

**Table 4 jpm-13-00073-t004:** Significant differences in linguistic dimensions between groups of women who undergo ART treatments for different infertility causes.

	Unknown Cause *n* = 63		Female Causes *n* = 123		Male Causes *n* = 62		Both Partner Causes *n* = 45		F	*p*
	M	SD	M	SD	M	SD	M	SD		
Anger	0.66	1.94	0.11	0.54	0.19	0.82	0.10	0.45	1.456	0.005
Inhibition	0.14	0.45	0.28	0.98	1.22	3.66	0.16	0.47	5.055	0.002
Body states and symptoms	0.25	0.67	0.26	0.80	0.72	1.60	0.51	1.73	2.611	0.049

## Data Availability

The data that support the findings of this study are available on request from the corresponding author [S.M.].

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
