# Peer review of "Toward a Personalized Psychological Counseling Service in Assisted Reproductive Technology Centers: A Qualitative Analysis of Couples’ Needs"

_jpm, 2022, doi:10.3390/jpm13010073_

Round 1

Reviewer 1 Report

Journal of Personalised Medicine (MDPI)_ 2083171

“Towards a personalised psychological counselling service in Assisted Reproductive Technology centres: a qualitative analysis of couples' needs”

In the present work the authors investigated the psychological impact or ART procedures in women through a methodology that involved the objective analysis the answers to questionnaire. Considering the importance and the social, physiological and conjugal impact of the ART procedures the study has relevant and considerable interest. The methodology and approach are adequate and correct. Hopefully, the researchers reached a considerable participation from patients allowing a significant sample dimension.

I have minor points.

Line 55. Besides the reasons that authors cited for the high dropout percentage, it is necessary consider that ART procedures are actually quite expensive, as not all ART procedures are financed by national health authorities, so this financial factor may constitute additional tension for the couple and a cause for dropout treatments.

In Material and Methods, which ART treatments were the participants of the study subjected to? Any descriptive statistic on them? At Line 120, clarify the participation of the men in the questionnaires. If 31 men participated in the questionnaire why to say that male participants were retrospectively excluded? (only the prospective were included?). The psychological impact in men could be addressed in future studies.

L 134 please correct repetition of “aimed”

Author Response

“Towards a personalised psychological counselling service in Assisted Reproductive Technology centres: a qualitative analysis of couples' needs”

In the present work the authors investigated the psychological impact or ART procedures in women through a methodology that involved the objective analysis the answers to questionnaire. Considering the importance and the social, physiological and conjugal impact of the ART procedures the study has relevant and considerable interest. The methodology and approach are adequate and correct. Hopefully, the researchers reached a considerable participation from patients allowing a significant sample dimension.

Authors: The Authors want to thank the Reviewer for the time devoted to review our manuscript and for the positive evaluation.

I have minor points.

Line 55. Besides the reasons that authors cited for the high dropout percentage, it is necessary consider that ART procedures are actually quite expensive, as not all ART procedures are financed by national health authorities, so this financial factor may constitute additional tension for the couple and a cause for dropout treatments.

Authors: According to Reviewer’s precious suggestion the importance of economic burden on ART treatments drop-out have been more clearly emphasized.

“Moreover, undergoing an Assisted Reproductive Technology (ART) treatment for fertility issues can represent a further element of fatigue and stress for these couples, since medical procedures could be long, demanding, economically burdensome, and invasive [8-10]….”

“The impact of diagnosis and medical procedure, both on psychological, physical and economic dimensions, seems to contribute to the high rates of treatment dropout, with an estimated 30% of couples discontinuing treatments”

In Material and Methods, which ART treatments were the participants of the study subjected to? Any descriptive statistic on them?

Authors: The Authors want to thank the Reviewer for the opportunity to better explore this point. A sentence regarding to which ART treatments the participants of the study were subjected to was added in the Participants section.

“The 93.5% of the women participating in the study (n=274) undergo In Vitro Fertiliza-tion/Intracytoplasmic Sperm Injection (IVF/ICSI, both second level technique based on in vitro fertilization with embryos transfer) whereas the 6.5% undergo Intrauterine In-semination (IUI, a first level technique).”

Considering the low rate of first level technique this point has been inserted in the limit section.

“Moreover, the rate of women undergoing first ART first level techniques was limited (6.5%) and in future studies this element should be considered.”

At Line 120, clarify the participation of the men in the questionnaires. If 31 men participated in the questionnaire why to say that male participants were retrospectively excluded? (only the prospective were included?). The psychological impact in men could be addressed in future studies.

Authors: The Authors want to thank the Reviewer for the opportunity to clarify this point. The participation to the study was open to both female and male patients but since the limited participation rate of males we decided to not include the male participants in the data analysis that therefore have been excluded. The term retrospectively was removed. The sentence has been reworded.

“A total of 324 patients undergoing ART treatment participated in the study (293 females 31 males) and completed the online questionnaire. The male participants were excluded from the data analysis since the reduced male participation do not allow a homogenous gender distribution and the generalizability of the results to the male population”.

L 134 please correct repetition of “aimed”

Authors: The Authors want to thank the Reviewer, the repetition of word aimed has been delated.

Reviewer 2 Report

Thank you for the opportunity to review the paper entitled “Towards a personalised psychological counselling service in 2 Assisted Reproductive Technology centres: a qualitative analy-3 sis of couples' needs.”

The research theme is really important.

To publish this article, please consider several points described below.

1. Aim

The aim of the study does not seem clear enough.

The authors may want to describe what kind of characteristics should be explored.

2. Analysis

I could not understand the reason why the authors chose one-way ANOVA, instead of t-test, when they compare the mean value between two groups.

Besides, I’m not sure whether parametric analyses were appropriate, since distributions were not shown.

3. Analysis

I’m not sure whether the number of each characteristic can be truly reflected by the status of ART or demographic situations. Regardless of such variables, those who wanted to disclose their emotion should tend to write more, compared to others. There should be considerable bias.

4. Results and Discussion

As Methods seems to produce substantial bias, the authors may want to interpret the data as such.

5. There are typos and grammatical errors in the manuscript. Please review the article wholly and revise them.

For example,

L.134 Aimed aimed...

L.138 you feel could...

Author Response

Thank you for the opportunity to review the paper entitled “Towards a personalised psychological counselling service in 2 Assisted Reproductive Technology centres: a qualitative analysis of couples' needs.”

The research theme is really important. To publish this article, please consider several points described below.

Authors: The Authors want to thank the Reviewer for the time devoted to review our manuscript and for the positive evaluation of the thematic.

  1. Aim

The aim of the study does not seem clear enough.

Authors: The Authors want to thank the Reviewer for this suggestion offering the opportunity to clarify this point.  The aim has been clarified both in the abstract and in the introduction section.

Abstract: “This study aims to explore the emotions related to the infertility and to the Assisted Reproductive Technology (ART) procedure for which patients consider useful a psychological support.” 

Introduction: “This study aims to explore the linguistic characteristics of texts written by infertile women undergoing ART treatments for fertility problems regarding their emotions related to the infertility and to ART procedure for which patients consider useful a psychological support”

  1. Analysis

I could not understand the reason why the authors chose one-way ANOVA, instead of t-test, when they compare the mean value between two groups. Besides, I’m not sure whether parametric analyses were appropriate, since distributions were not shown.

Authors: The Authors want to thank the Reviewer for this suggestion. Since most of the linguistic measures did not follow normal distribution, differences between sub-groups of women in the linguistic measures were evaluated with the use of the Mann-Whitney U and Kruskal-Wallis non-parametric tests for in-dependent samples, to compare two sub-groups or more than two sub-groups respectively. The statistical analysis section, the results section and discussion section were deeply revised according to this.

“Statistical Analysis

A statistical analysis was conducted using the Statistical Package for Social Science (SPSS) Version 25 for Windows (IBM, Armonk, NY, United States). The women who participated in this study were divided in different sub-groups according to the specific socio-anamnestic characteristics considered (age under/over 40 years, the first ART attempt versus the second or another attempt, ART with/without gamete donation, and specific cause of infertility). Since most of the linguistic measures did not follow normal distribution, differences between sub-groups of women were evaluated with the use of the Mann-Whitney U and Kruskal-Wallis non-parametric tests for independent samples, to compare two sub-groups or more than two sub-groups respectively. Furthermore, when Kruskal–Wallis tests was used the Mann–Whitney test was performed as post hoc test to determine the specific differences between sub-groups and Bonferroni correction for multiple comparisons was applied. A p <0.05 was considered significant.

“Results

Table 3 shows the results of the Mann-Whitney U non-parametric tests run. In regards to the variable age, the group of women who were 40 years or younger scored higher as regards the use of words related to negative emotions, social process, communication process, future actions, exclusive terminology (such as: but, except, without), and leisure activity, compared to women who were 40 years or older. Regarding the variable of ART attempts, the group of women undergoing their first attempt showed a lower use of words expressing negations, optimism and energy as well as a greater use of words expressing leisure activity, compared to women undergoing their second or further attempt (see Table 3). Regarding the variable of ART treatment with/without gamete donation, the women undergoing the medical procedure with gamete donation showed a lower word count and words per sentence, a lower use of words relate to sex and sensuality, to the body and first person plural pronouns, well as a greater use of words related to death and dying compared to women using their own gamete.

Regarding the specific cause of infertility, several significances emerged for the use of words related to feelings of anger, inhibition, as well as for the use of first person singular pronouns (see Table 4). Multiple comparison post-hoc analysis with Bonferroni correction showed that the group of women who underwent the ART procedure without a clear cause of infertility reported a greater use of words reflecting feelings of anger, compared to the women who underwent ART for female (U= 23.595; p=.004), male (U=21.098 p=.049) or both partners’ cause (U=23.330; p=.044). As regards the differences in the use of words related to inhibition and block processes post-hoc comparisons did not show significances. However, the group of women who underwent ART for male cause showed a tendency to the significance for a greater use of words related to inhibition and block processes than the groups of women reporting an unexplained or female infertility cause (both a p=.06). The women who underwent ART for female cause show a greater use of first person singular pronouns than women with a male (U= 33.320; p=.035).”

  1. Analysis

I’m not sure whether the number of each characteristic can be truly reflected by the status of ART or demographic situations. Regardless of such variables, those who wanted to disclose their emotion should tend to write more, compared to others. There should be considerable bias.

Authors: The authors are not sure that they have correctly understood the reviewer's comment. The variables age over 40, previous attempts, ART with gamete donation and specific infertility cause have been chosen in accordance with the international literature highlighting their negative impact on psychological and emotional wellbeing of people involved. Therefore, it may be hypothesized that these elements may influence the way women talk about their experience. Moreover, the length of texts is explored in each comparison but only in some comparisons significant differences emerged. Furthermore, Authors do not think that a greater length of texts was necessary associated to a greater self-disclosure about emotional needs associated to the experience faced.

  1. Results and Discussion

As Methods seems to produce substantial bias, the authors may want to interpret the data as such.

Authors: According to reviewer’s suggestion the statistical analysis and results were deeply revised. Moreover, a more cautious language has been used to interpret data. 

There are typos and grammatical errors in the manuscript. Please review the article wholly and revise them.

Authors: Authors corrected the typos and grammatical errors in the manuscript.